

# Wind farm layout optimization with alignment constraints

Paul Malisani[1], Tristan Bartement[2], and Pauline Bozonnet[3]

[1]IFP Energies nouvelles, Applied Mathematics Department, 1 et 4 avenue de Bois-Préau, 92852 Rueil-Malmaison, France
[2]IFP Energies nouvelles, Scientific Computing Department, 1 et 4 avenue de Bois-Préau, 92852 Rueil-Malmaison, France
[3]GreenWITS, Rond-Point de l'échangeur, les Levées, 69360 Solaize, France

**Correspondence:** Paul Malisani (paul.malisani@ifpen.fr)

**Abstract.** Wind farm layout optimization involves placing wind turbines in a defined domain to minimize the expected production losses due to wake effects within the wind farm. Because of navigational regulations, tenders for offshore wind farms often impose so-called alignment constraints, i.e., wind turbines must be located at the intersections of a parallelogram-made grid. The shape and orientation of these parallelograms are to be optimally determined to minimize wake losses. To the authors' best
knowledge and despite its practical interest, the wind farm layout optimization problem under alignment constraints has not been investigated in the literature. The contributions of this paper are twofold, the first contribution is a dedicated optimization method to handle this problem, and the second contribution is to provide a challenging benchmark based on open data for layout optimization with alignment constraints.

## 1 Introduction

Selecting a proper layout is an important task when building a wind farm. A layout far from optimal is prone to significant loss of expected Annual Energy Production (AEP) due to wake effects within the farm. Having an optimized method of turbine placement in a given area helps to maximize the energy production over the life span of the wind farm. In its full generality, the problem of optimizing a wind farm layout is a complex one for several reasons. The first is that computing a given farm's mean annual energy production is numerically complex, i.e., evaluating the optimization problem's objective function
is computationally time-consuming. The second difficulty is that the problem is not convex, i.e., neither the objective function nor the minimization set are convex. As a result, the problem of wind farm layout optimization requires developing dedicated optimization tools. Wind farm optimization has been the subject of much scientific research, see for example, Herbert-Acero et al. (2014) for a review of such methods. Let us now focus on recent contributions to the field. In Quick et al. (2023), the authors develop a stochastic gradient-based method for wind farm optimization. The presented algorithm is developed for
circular or square domains and could be easily extended to convex domains but not to non-convex or non-connected ones. In Kumar and Sharma (2023), the authors use a so-called teaching-learning-based algorithm to also solve the wind farm layout optimization problem for a circular domain. In Liang and Liu (2023), the authors use genetics and particle swarm algorithms to solve the problem on a square domain. In Fischetti and Fischetti (2022), the authors propose a Mixed Integer Linear Programming model to solve turbine placement and cable routing optimization problems. In Kunakote et al. (2022),
the authors compare 12 meta-heuristic methods for wind farm layout optimization. The presented methods do not rely on





any assumption on the shape of the admissible domain. However, the method relies on a very coarse discretization of the domain and can be numerically intractable using a finer one. In Thomas et al. (2023), the authors compare eight wind farm optimization methods and provide a benchmark case study for comparing algorithm performances. The proposed benchmark is highly complex since the admissible domain is neither convex nor connected. However, no alignment constraints are taken into account in this contribution. Finally, in Stanley and Ning (2019), the authors propose a so-called inner-grid wind farm layout parameterization that satisfies strong alignment constraints. However, the authors assume a one-to-one correspondence between this parameterization and the layout configuration, dramatically reducing the degree of freedom and potentially leading to far-from-optimal solutions. In fact, despite the extensive literature on wind farm optimization, to the best of our knowledge, no other method than the latter can handle turbine alignment constraints. By alignment constraint, we mean that we place the turbines on the intersections of a regular grid made of parallelograms whose shape and orientation are to be determined. Far from being just an academic question, turbine alignment constraints are, in practice, often imposed on developers by maritime authorities to secure the navigation of boats within the wind farm. The contribution of this paper is to provide an optimization algorithm for wind farm layout optimization that is able to handle alignment constraints. Mathematically, these alignment constraints make the wind farm layout optimization problem a mixed integer non-linear programming (MINLP). The integer variables are the positions of the turbines, which belong to the finite set of grid intersections located in the admissible domain. The continuous variables are the grid parameters, that is to say the size and orientation of the grid's unit parallelogram. The non-linearity mainly stems from the wind farm's Annual Energy Production (AEP) as a function of the optimization parameters. These problems are generally extremely difficult to solve. As detailed in Burer and Letchford (2012), solving algorithms for non-convex MINLP falls into two different categories: exact methods and heuristics-based methods. Exact methods often rely on Branch and Bound methods Papadimitriou and Steiglitz (1998) or separation properties of the objective function. Heuristics methods include tabu research Exler et al. (2008), particle swarm algorithms Yiqing et al. (2007); Young et al. (2007), genetic algorithms Schlüter et al. (2009) or local search methods Liberti et al. (2011). The strategy adopted in this paper is to adapt the DEBO method the authors developed in Thomas et al. (2023) to the problem at hand. This method is a local-search-based method coupled with an optimization-parameters-set exploration heuristic. The paper is organized as follows. In section 2, we introduce useful notations and definitions. In section 3, we describe the aligned-layout optimization problem; that is to say, we present the objective function, the constraints, and the optimization variables. We fully describe the optimization algorithm in section 4. In section 5, we take up the benchmark presented in Thomas et al. (2023) and add the alignment constraints, and we conduct a thorough study on the setting of the optimization algorithm hyper-parameters. In section 6, we illustrate the impact of the alignment-grid parameters exploration method on the AEP and prove that an efficient exploration method yields a strong improvement of the AEP. Finally, in section 7, we give the conclusions of this work and draw up research perspectives on the subject.

## 2 Notations and definitions

Throughout the paper we will use recurrently the following notations





- $\mathbb{R}, \mathbb{R}_+$ denote respectively the set of real numbers and the set of non negative real numbers.

- $\mathbb{Z}, \mathbb{Z}_*$ denote respectively the set of integers and the set of non zero integers

- $\Omega \in \mathbb{R}^2$: Two dimensional domain where turbines can be planted

- $N_{\max}$: Maximal number of turbines to be placed within the admissible domain $E$

- $\text{turb}_{\text{diam}}$: Turbine diameter

- $D_{\min}$: Minimal distance between turbines

- $D_{\max}$: Maximal distance between turbines

- $w_s$: Wind speed

- $w_d$: Wind direction

- $\top$: Logical True

- $\bot$: Logical False

- $\neg$: Logical negation

- $\wedge$: Logical conjonction (and)

- $\vee$: Logical disjonction (or)

**Definition 1** (Wind farm). *A capital bold character associated with a subscript such as $\mathbf{F}_n$ denotes a n-turbines wind farm. Mathematically $\mathbf{F}_n$ is a function satisfying:*

$$\mathbf{F}_n : \{1, \ldots, n\} \ni k \mapsto (x_k \ y_k)^\top \in \mathbb{R}^2 \tag{1}$$

*where $(x_k \ y_k)^\top$ is the position of the $k^{th}$-turbine. Let $\mathbf{F}_n$ and let $(x \ y)^\top \in \mathbb{R}^2$, we denote $\mathbf{H}_{n+1} := \mathbf{F}_n \oplus (x \ y)$ the wind farm defined as follows*

$$\mathbf{H}_{n+1}(k) := \begin{cases} \mathbf{F}_n(k) & \text{if } k \leq n \\ (x \ y)^\top & \text{if } k = n+1 \end{cases} \tag{2}$$

**Definition 2** (Wind farm Power Production). *We denote $\mathcal{P} : (\mathbf{F}_n, w_s, w_d) \mapsto \mathbb{R}_+$ the power production of the wind farm $\mathbf{F}_n$*
*for a wind speed $w_s$ and a wind direction $w_d$.*

**Definition 3** (Wind random variable). *Let $(E, \mathbb{P})$ be a probability space. We denote $\mathrm{W} : E \mapsto Z := \mathbb{R}^+ \times [0, 2\pi)$ the wind random variable which associates a random event, denoted $e \in E$, to a wind configuration $(w_s \ w_d) \in Z$, that is to say, $\mathrm{W}(e) =$*



$(w_s \ w_d)$. *We also denote* $\mathbb{P}_W$ *the probability measure on* $Z$ *associated with the random variable* $W$. *Finally, we denote* $\mathbb{E}_W$ *the expectation with respect to the probability* $\mathbb{P}_W$ *defined as*

$$\mathbb{E}_W(f) := \int_Z f(w_s, w_d) d\mathbb{P}_W(w_s, w_d) \tag{3}$$

*for all measurable function* $f : Z \mapsto \mathbb{R}$.

## 3 Optimization model for layout optimization with alignment constraints

The optimization problem we are interested in consists of optimizing the grid configuration and the turbine placement on the intersections of this grid. This optimization problem is a non-linear mixed integer programming problem, a class of problems known to be challenging to solve. In this section, we describe the parameterization of our problem.

### 3.1 Grid parameterization

To write the optimization problem, we parameterize the grid using 6 parameters $(r_1, r_2, \theta_1, \theta_2, v_x, v_y)$ as represented on fig. 1. Using this parameterization we define the change-of-basis matrix from the canonical basis denoted $\mathcal{B}_0$ to the grid basis denoted $\mathcal{B}(\theta_1, \theta_2, r_1, r_2)$ as follows

$$P_{\mathcal{B}_0}^{\mathcal{B}(\theta_1,\theta_2,r_1,r_2)} = \begin{pmatrix} r_1\cos(\theta_1) & r_2\cos(\theta_2) \\ r_1\sin(\theta_1) & r_2\sin(\theta_2) \end{pmatrix} \tag{4}$$

In the grid-basis coordinates, any intersection point $p$ writes as follows:

$$p := \begin{pmatrix} k_1 \\ k_2 \end{pmatrix} + \left( P_{\mathcal{B}_0}^{\mathcal{B}(\theta_1,\theta_2,r_1,r_2)} \right)^{-1} \begin{pmatrix} v_x \\ v_y \end{pmatrix} \quad k_1, k_2 \in \mathbb{Z}$$

This parameterization in grid basis is, in turn, equivalent to

$$p = \begin{pmatrix} k_1 \\ k_2 \end{pmatrix} + \begin{pmatrix} \Delta_1 \\ \Delta_2 \end{pmatrix} \quad k_1, k_2 \in \mathbb{Z} \text{ and } \Delta_1, \Delta_2 \in [0, 1) \tag{5}$$

### 3.2 Layout parameterization

Using the grid parameterization described in section 3.1, an aligned layout has its turbines located on the intersections of the grid which writes

$$\mathbf{F}[\mathcal{B}]_n(i) : \{1, \ldots, n\} \ni i \mapsto \begin{pmatrix} k_1^i & k_2^i \end{pmatrix}^\top + \begin{pmatrix} \Delta_1 & \Delta_2 \end{pmatrix}^\top, \quad k_1^i, k_2^i \in \mathbb{Z}; \ \Delta_1, \Delta_2 \in [0, 1) \tag{6}$$

The corresponding wind farm in canonical coordinates $\mathbf{F}_n$ thus writes

$$\mathbf{F}_n(i) = P_{\mathcal{B}_0}^{\mathcal{B}(\theta_1,\theta_2,r_1,r_2)} \mathbf{F}[\mathcal{B}]_n(i), \ i = 1, \ldots, n \tag{7}$$





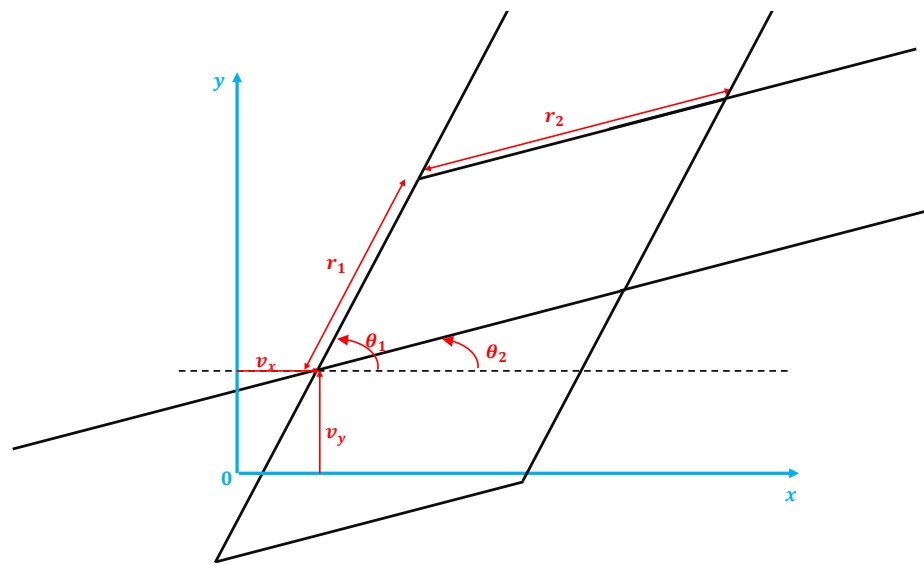

**Figure 1.** Grid parameterization for aligned layout

### 3.3 Optimization problem

We are now ready to write the general wind farm layout optimization problem with alignment constraints. This optimization problem consists of maximizing the AEP, which, in turn, is equivalent to maximizing the expected power production of the wind farm with respect to the wind random variable as defined in eq. (3). This writes

$$\max_{(k_1^i, k_2^i)_{i=1,\dots,N_{\max}}, \Delta_1, \Delta_2, r_1, r_2, \theta_1, \theta_2} \mathbb{E}_{\mathrm{W}} \left[ \mathcal{P}(\mathbf{F}[\mathcal{B}]_{N_{\max}}, w_s, w_d) \right] \tag{8}$$

under the following constraints

$$k_1^i, k_2^i \in \mathbb{Z}; \ i = 1, \dots, N_{\max} \tag{9}$$

$$\left| k_1^i - k_1^j \right| + \left| k_2^i - k_2^j \right| \geq 1; \ \forall i \neq j \tag{10}$$

$$\Delta_1, \Delta_2 \in [0, 1) \tag{11}$$

$$\mathbf{F}_{N_{\max}}(i) \in \Omega; \ i = 1, \dots, N_{\max} \tag{12}$$

$$\theta_1 \in \left( -\frac{\pi}{2}, \frac{\pi}{2} \right] \tag{13}$$

$$\theta_2 \in \left[ -\frac{\pi}{2}, \theta_1 \right) \tag{14}$$

$$D_{\min} \leq \min_{z \in \mathbb{Z}_*^2} \left\| P_{\mathcal{B}_0}^{\mathcal{B}(\theta_1, \theta_2, r_1, r_2)} z \right\| \tag{15}$$

Equations (9) to (11) ensure that the turbines are located at the intersections of the grid defined by the parameters $(r_1, r_2, \theta_1, \theta_2, v_x, v_y)$ and thus, that the turbines are aligned along the directions $\theta_1$ and $\theta_2$. The constraint from eq. (12) guar-



antees that the turbines are located in the admissible domain. Constraints defined in eqs. (13) and (14) allow to generate all possible parallelogram-based grids. Finally, the constraint from eq. (15) guarantees that the minimal distance between two turbines is greater than $D_{\min}$.

## 4 Solving Algorithm

### 4.1 General description of the proposed solving algorithm

The method presented in this paper belongs to the category of heuristic-based methods and consists in the following steps.

1. The first step consists of computing the set $R_{1,2}$ of parameters $(r_1, r_2)$ by discretization of the space $[D_{\min}, D_{\max}]^2$ using a grid size of $\Delta r$.

2. The second step consists in reducing the size of the angle-search space. To do so, for each couple $(r_1, r_2) \in R_{1,2}$, we discretize the search space $\left(-\frac{\pi}{2}, \frac{\pi}{2}\right] \times \left[-\frac{\pi}{2}, \theta_1\right)$ using a discretization size of $\Delta\theta$. Then, for each grid configuration $(r_1, r_2, \theta_1^k, \theta_2^k)_k$ satisfying eq. (15) we compute the AEP of an elementary 4 turbines wind farm. Then, we store the $N_\theta$ best angle configurations $(\theta_1, \theta_2)$ in a angle set $\Theta$. This latter set $\Theta$ is the reduced angle-search space.

3. Then, we compute a set of grid configurations $(r_1, r_2, \theta_1, \theta_2)$ denoted grids defined as grids := $\{(r_1, r_2, \theta_1, \theta_2) : (r_1, r_2) \in R_{1,2}, (\theta_1, \theta_2) \in \Theta, eq. (15) \text{ holds}\}$

4. For each explored shape configuration $(r_1, r_2, \theta_1, \theta_2)$, compute an optimal layout using a greedy algorithm for placing the $N_{\max}$-turbines on the intersections of the grid and using a local search optimization method to move the turbines on the intersections. The sequence of greedy initialization followed by a local search method has already been proved efficient for wind farm layout optimization without alignment constraints, see the DEBO algorithm from Thomas et al. (2023).

### 4.2 Angle search space reduction and grid configuration selection

In this section, we describe the first part of the algorithm which consists in finding a set of grid configurations $(r_1, r_2, \theta_1, \theta_2)$ of reasonable size and to perform a complete wind farm layout optimization for each element of this set. To do so, we first compute the set $R_{1,2}$ of parameters $(r_1, r_2)$ by discretizing the space $[D_{\min}, D_{\max}]^2$ using a discretization of size $\Delta r$. Then, for each $(r_1, r_2) \in R_{1,2}$, discretize the following search space

$$S := \left\{ (\theta_1, \theta_2) \in \left[-\frac{\pi}{2}, \frac{\pi}{2}\right]^2 : \theta_1 \geq -\frac{\pi}{2} + \Delta\theta, \, \theta_2 \leq \theta_1 - \Delta\theta \right\} \tag{16}$$

with an angle discretization parameter $\Delta\theta$. We denote $S_D$ this discrete search space. Then, for each $(r_1, r_2) \in R_{1,2}$, we define the corresponding angle search space $\Theta_{r_1, r_2}$ as follows

$$\Theta_{r_1, r_2} := \{(\theta_1, \theta_2) \in S_D : \text{eq. (15) holds}\} \tag{17}$$





Then, for each configuration $(r_1, r_2, \theta_1^k, \theta_2^k)_k$ with $(\theta_1^k, \theta_2^k) \in \Theta_{r_1, r_2}$ we compute the AEP of the elementary farms $(\mathbf{F}[\mathcal{B}(r_1, r_2, \theta_1^k, \theta_2^k)]_4)_k$

defined as follows

$$
\mathbf{F}[\mathcal{B}(r_1, r_2, \theta_1^k, \theta_2^k)]_4(i) := \begin{cases} (0\ 0)^\top & \text{if } i = 1 \\ (1\ 0)^\top & \text{if } i = 2 \\ (0\ 1)^\top & \text{if } i = 3 \\ (1\ 1)^\top & \text{if } i = 4 \end{cases} \tag{18}
$$

and sort the couples $(\theta_1^k, \theta_2^k)_i$ by decreasing order of AEP. Finally, for each $(r_1, r_2) \in R_{1,2}$, we store in the set $\Theta$ the best $N_\theta$ angles configuration $(\theta_1^k, \theta_2^k) \in \Theta_{r_1, r_2}$. Then, the continuous variables search-space denoted $\mathrm{grids}$ consists in all the combinations of the $(r_1, r_2) \in R_{1,2}$ explored with all the angles configuration from $\Theta$, i.e.

$\mathrm{grids} := \{(r_1, r_2, \theta_1, \theta_2) : (r_1, r_2) \in R_{1,2}, (\theta_1, \theta_2) \in \Theta : eq.\ (15)\ \text{holds}\}$ \hfill (19)

The corresponding algorithm in pseudo-code is described in algorithm A1, algorithm A2, algorithm A3, algorithm A4.

### 4.3   Compute intersections for each grid configuration

This part of the algorithm consists of traversing $\mathrm{grids}(k)$ and, for each configuration $(r_1^k, r_2^k, \theta_1^k, \theta_2^k)$, calculating the maximum number of intersections located in the admissible domain $\Omega$ and their positions. If this set of intersections has more than $N_{\max}$

elements and if all intersections are $D_{\min}$-apart from each other, this set of intersections is stored in a set of set-of-intersections that we denote $\mathrm{intersections\_sets}$. The corresponding algorithm is written in pseudo-code in algorithm A5.

### 4.4   Optimize turbines placement

#### 4.4.1   Greedy Initialization

Given a grid configuration $(r_1, r_2, \theta_1, \theta_2, \Delta_1, \Delta_2)$, a greedy algorithm is used to sequentially place $N_{\max}$ turbines on the

admissible intersections. This algorithm consists in sequentially placing the turbines on the best possible empty intersection, in the sense of AEP maximization, until $N_{\max}$ turbines are placed. The corresponding algorithm in pseudo-code is given in algorithm A6.

#### 4.4.2   Local Search

This part of the algorithm sequentially moves each turbine in random order from its current intersection to a free one if it

provides a strict increase in AEP. The algorithm stops when a complete course of all the turbines has been made without a single one being moved. When the number of intersections in the admissible domain is much bigger than $N_{\max}$, one can explore a subset of the free intersections. For example, one can explore the $p$ closest intersections from the turbine to be moved or select $p$ random free intersections. In this case, the size of the subset to explore and its definition, $p$, are user-defined parameters. The algorithm in pseudo-code is given in algorithm A7.





### 4.4.3 Turbine placement optimization

Finally, given a set of intersections, the optimization algorithm for optimal turbine placement consists of using sequentially the greedy initialization and the local search algorithm as described in pseudo-code in algorithm A9.

### 4.5 Complete Algorithm

Finally, the complete algorithm includes all the blocks described in sections 4.3 and 4.4. The corresponding pseudo-code is displayed in algorithm A10. As one can see on lines 2, and 6 from algorithm A10, large parts of the proposed algorithm can be run in parallel.

## 5 Numerical Examples

### 5.1 Problem presentation

For this numerical example, we use the same case study as in Thomas et al. (2023), whose data are available in Baker et al. (2021). This case study was created within the International Energy Association (IEA) Wind Task 37, and is based on the Borssele III and IV wind farms. Of particular interest in this case study is the presence of five disconnected boundary regions and concave boundary features, as shown in fig. 2. The turbines are 10 MW machines with 198 m rotor diameters based on the IEA 10 MW reference wind turbine (Bortolotti et al. (2019)). For the AEP computation, we also use the same algorithm as in Thomas et al. (2023). This method is based on a simple Gaussian wake model based on Bastankhah's Gaussian wake model (Bastankhah and Porté-Agel (2016)), and presented in the IEA case study 3 and 4 announcement documents (Baker et al. (2021)), to calculate wind speeds at each turbine in the wind farm. However, any other AEP computation software, such as FLORIS, can be used with the presented algorithm as long as the computation time of the AEP is fast enough. Indeed, our optimization algorithm requires a large number of AEP evaluations.

### 5.2 Influence of the hyper-parameters on the AEP and the computation time

The optimization procedure described in section 4 requires setting 5 hyper-parameters $D_{\min}$, $D_{\max}$, $\Delta\theta$, $N_\theta$, and $\Delta r$. The turbine's manufacturer usually sets $D_{\min}$ at a fixed value. In this example, we set $D_{\min} = 2\text{turb}_{\text{diam}}$. The parameter $D_{\max}$ must be chosen large enough to allow a good exploration range for the grid parameters $(r_1, r_2)$. However, setting $D_{\max}$ with a significant value generates grids with a number of admissible intersections smaller than the number of turbines to be placed, i.e., in generating non-admissible layouts. Therefore, we have set $D_{\max} = 6\text{turb}_{\text{diam}}$. The parameter $\Delta\theta$ should be chosen to the minimal value such that the wake model used to compute the AEP is valid. For this example, we set $\Delta\theta = 1°$. The remaining hyper-parameters, $N_\theta$, and $\Delta r$, dramatically affect the optimization procedure regarding AEP value and computation time. Indeed, as explained in section 4.2, the larger $N_\theta$, the larger the reduced angle-set denoted $\Theta$, and the smaller $\Delta r$, the larger the set $R_{1,2}$. The number of configurations to optimize being the product of the cardinal of the sets $\Theta$ and $R_{1,2}$, the larger these sets, the longer the computation time. However, the more configuration to optimize, the greater the AEP. Therefore,



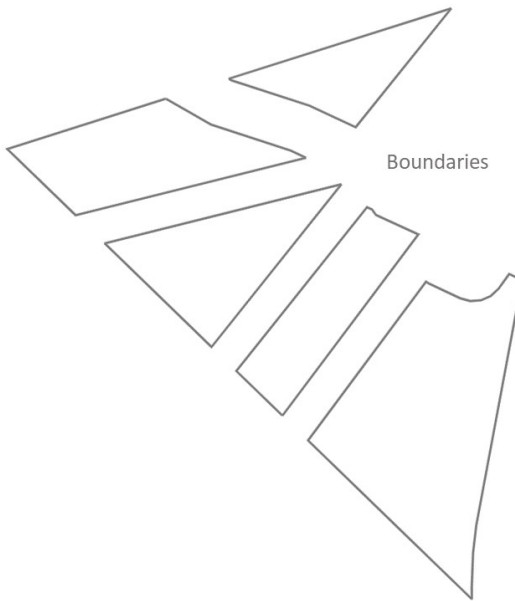

**Figure 2.** An overhead view of the boundaries of the admissible domain $\Omega$

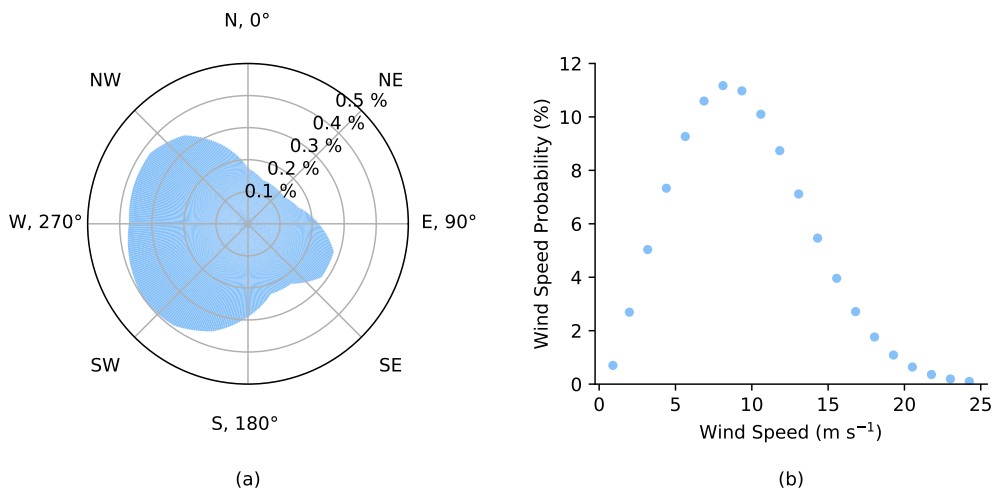

**Figure 3.** This figure, reproduced from Thomas et al. (2023), displays the full wind resource used for evaluating the final wind farm layouts. (a) The wind direction probability (360 bins). (b) A representative wind speed probability distribution (20 bins).

any layout optimization needs to make a trade-off between computation time and size of the set of configuration to optimize. In this section, we will show the effect of the parameters $N_\theta$ and $\Delta r$ on the optimal AEP and the computation time, and give the user some guidelines to set these parameters. To do so, we run algorithm A10 for all possible configurations of the





hyper-parameters valued in their respective value-set given in table 1, for wind farm sizes of 81, 100, 150, and 250 turbines respectively. The numerical results of these optimizations are gathered in table 2 and illustrated in figs. 4 and 5. On fig. 4,

| $D_{\min}$ | $D_{\max}$ | $\Delta\theta$ | $N_\theta$ | $\Delta r$ |
|---|---|---|---|---|
| $\{2\,\mathrm{turb_{diam}}\}$ | $\{6\,\mathrm{turb_{diam}}\}$ | $\{1°\}$ | $\{1,5,10\}$ | $\left\{\frac{\mathrm{turb_{diam}}}{2}, \mathrm{turb_{diam}}, 2\,\mathrm{turb_{diam}}\right\}$ |

**Table 1.** Value set of each hyper-parameter

one can see that the expected power per turbine[1] is growing with respect to $N_\theta$ and decreasing with respect to $\Delta r$. Also, when $N_\theta \geq 5$ and $\Delta r \leq 1\mathrm{turb_{diam}}$, the expected powers per turbine are similar whatever the value of these hyper parameters. However, as illustrated on fig. 5, the execution time is strongly increasing with respect to $N_\theta$ and strongly decreasing with respect to $\Delta r$. Therefore, if algorithm A10 is run using an AEP computation method more precise and computationally more expensive than the one we used, (see Baker et al. (2021); Thomas et al. (2023)), keeping $N_\theta$ reasonably small ($\approx 5$) and $\Delta r$

reasonably large ($\approx 1$) should enable the solving algorithm to find a well-performing layout in a reasonable execution time. In addition, as illustrated in table 1, the solving algorithm often sets the optimal shape parameters $(r_1, r_2)$ to their minimal authorized values. This behavior indicates that the method generates grids with many intersections and, thus, a large degree of freedom for the local search part of the algorithm. The larger the degree of freedom for the local-search algorithm, the better the solution. The best layout for each wind farm size is displayed on fig. 6 and the optimal parameters $(r_1, r_2, \theta_1, \theta_2)$, the optimal

AEP, the optimal expected power per turbine, and the execution time are displayed on table 3.

## 6   Exploration method's impact on the AEP

The optimization layout algorithm presented in this paper relies on the discrete exploration of the space of shape parameters $(r_1, r_2, \theta_1, \theta_2) \in [D_{\min}, D_{\max}]^2 \times [-\pi/2, \pi/2]^2$. Despite the angles' search-space reduction technique presented in section 4.2 and algorithm A2, exploring this space using fine discretization is numerically too demanding. Unfortunately, the performance

of the optimization depends on the size of the discretization step; the smaller this size, the higher the AEP. Therefore, there is a strong incentive to develop heuristic methods to explore the shape parameters space other than by using the angle search space reduction associated with a brute force-like exploration method. In this section, we present optimization results using such a heuristic to provide a benchmark for an aligned optimization algorithm. Unfortunately, for industrial confidentiality reasons, we do not describe its principle and only focus on the improvement in terms of AEP. Again, we have run the optimization

procedure for wind farms of 81, 100, 150, and 250 turbines. The results in terms of optimal shape parameters, AEP, expected power per turbine, and wake losses are summarized in table 4, and the optimal layouts are displayed in fig. 7. The optimal layout obtained using this heuristic exhibits the same behavior as those found in the previous section in terms of $r_1$ and $r_2$. Indeed, these parameters are systemically found to be equal to the lowest possible value. On the contrary, the optimal angles are not the same. One of the alignment directions is conserved ($\approx 18°$), but the other one is quite different even when taking

---

[1]For a $N_t$-turbines wind farm, the expected power per turbine is given by the formula $\mathrm{AEP(MWh)}/(8760 \times N_t)$ and allows for comparing wind farms of different sizes.

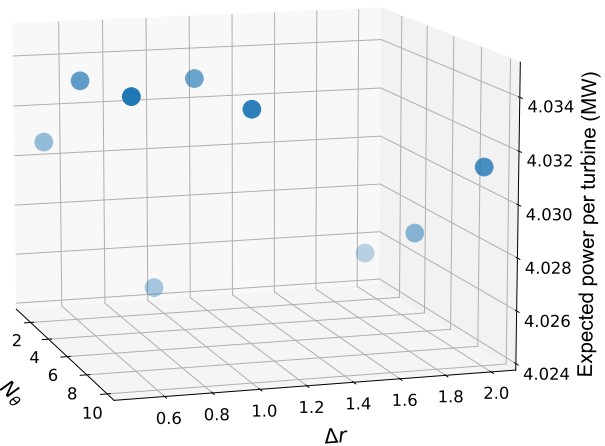

(a) Expected power per turbines for a 81-turbines wind farm as a function of $N_\theta$ and $\Delta r$

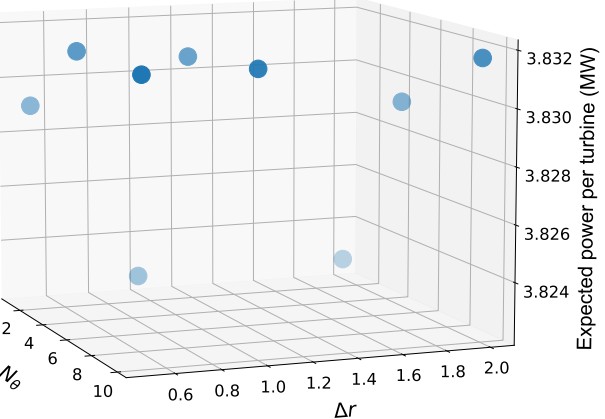

(b) Expected power per turbines for a 100-turbines wind farm as a function of $N_\theta$ and $\Delta r$

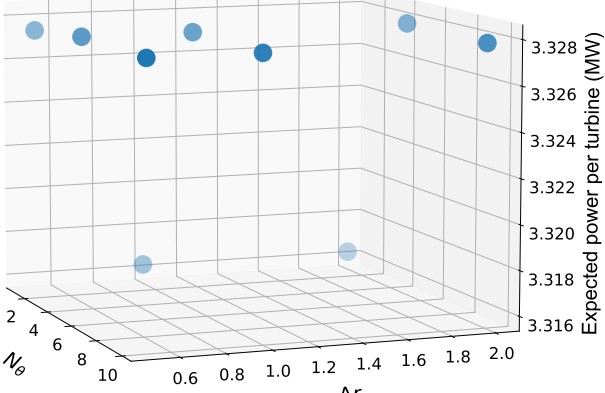

(c) Expected power per turbines for a 150-turbines wind farm as a function of $N_\theta$ and $\Delta r$

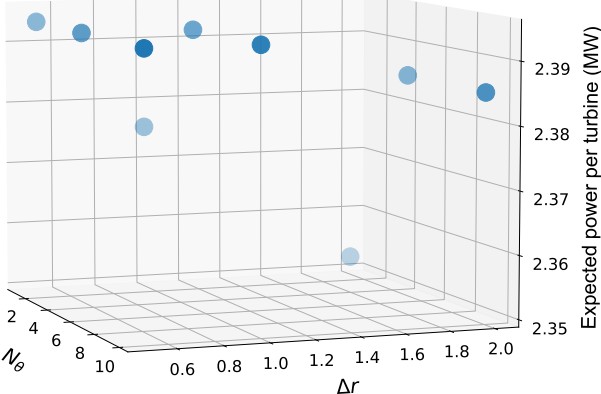

(d) Expected power per turbines for a 250-turbines wind farm as a function of $N_\theta$ and $\Delta r$

**Figure 4.** Expected power per turbine for 81, 100, 150, and 250 turbines wind farms as a function of the hyper-parameters $N_\theta$, and $\Delta r$.

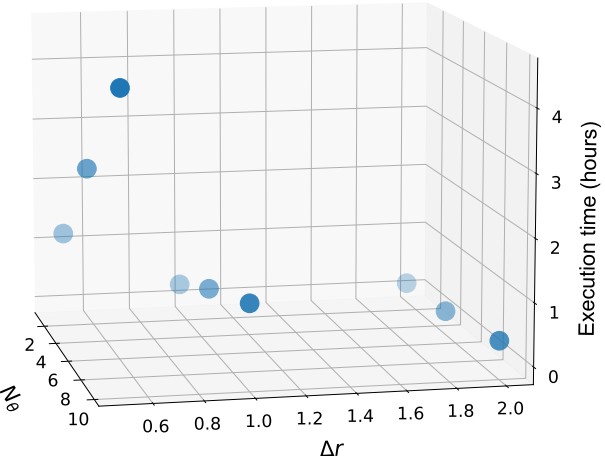

(a) Execution time of algorithm A10 as a function of the hyperparameters $N_\theta$, $\Delta r$ for a 81-turbines wind farm

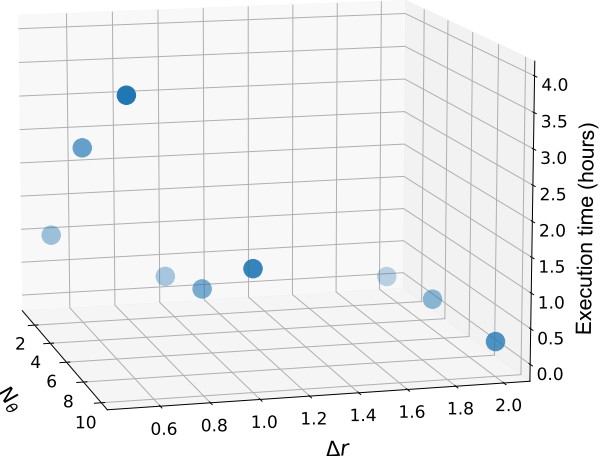

(b) Execution time of algorithm A10 as a function of the hyperparameters $N_\theta$, $\Delta r$ for a 100-turbines wind farm

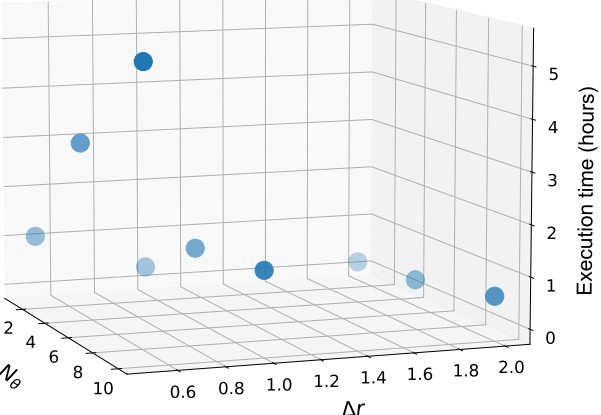

(c) Execution time of algorithm A10 as a function of the hyperparameters $N_\theta$, $\Delta r$ for a 150-turbines wind farm

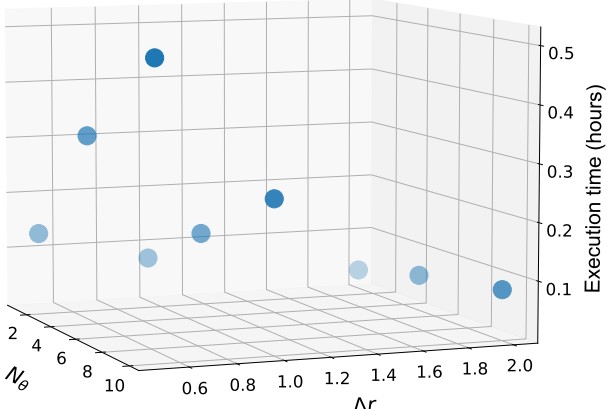

(d) Execution time of algorithm A10 as a function of the hyperparameters $N_\theta$, $\Delta r$ for a 250-turbines wind farm

**Figure 5.** Execution time of algorithm A10 for 81, 100, 150, and 250 turbines wind farms as a function of the hyper-parameters $N_\theta$, and $\Delta r$. All optimizations were run on a 12[th] Gen Intel(R) i7-12700H 2.30 GHz core.



(a) Best 81-turbines wind farm layout

(b) Best 100-turbines wind farm layout

(c) Best 150-turbines wind farm layout

(d) Best 250-turbines wind farm layout

**Figure 6.** Best layouts for wind farm sizes of 81, 100, 150, 250 turbines using algorithm A10.



into account the 180° periodicity of the angles. Concerning the AEP, using a heuristic to explore the space of shape parameters more efficiently allows for improvement. Interestingly, the percentage of AEP increase grows almost linearly concerning the wind farm size and reaches 1% for the larger one. This behavior stems from the decreasing degrees of freedom in the turbine's optimal placing problem as the wind farm size grows. Therefore, for larger farms, the efficiency of the shape parameters optimization algorithm is of greater importance than for smaller farms; thus, there is a more substantial improvement of AEP

for large farms when using a better exploration algorithm for the space of shape parameters. These results prove a strong interest in developing efficient heuristics to explore the space of shape parameters.

## 7 Conclusions

This work tackles the wind farm layout optimization problem with alignment constraints. We introduced a model of the corresponding optimization problem and adapted the DEBO algorithm from Thomas et al. (2023) to this new problem. The proposed

method is based on an exploration heuristic for computing the grid parameters and a local-search method to place the turbines on the grid's intersections optimally. We have shown that this method performs well on the benchmark of IEA Wind task 37 (Thomas (2022)) by outperforming the initial layout, even though the latter does not satisfy any alignment constraints and is potentially less prone to significant wake losses. Using this numerical example, we have also demonstrated the benefits of developing efficient heuristics for exploring the grid parameters. Indeed, using efficient heuristics allows for a better trade-

off between wake-losses reduction and computation time. Therefore, these heuristics can be used to find layouts with higher AEP or to use more precise and computationally demanding AEP models. Finally, a more efficient algorithm can enable the introduction of other optimization parameters or constraints, such as cable routing or shared mooring for floating farms.

*Data availability.* Provided physics model, turbines and boundary data are available at Thomas (2022), optimal layouts, AEPs, shape configurations, and intersections are available at Malisani (2024)



(a) Best 81-turbines wind farm layout

(b) Best 100-turbines wind farm layout

(c) Best 150-turbines wind farm layout

(d) Best 250-turbines wind farm layout

**Figure 7.** Best layouts for wind farm sizes of 81, 100, 150, 250 turbines using an efficient shape parameters exploration method and the same local-search algorithm.

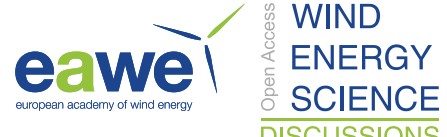

| Wind farm size | $\Delta r$ | $N_\theta$ | $(r_1, r_2, \theta_1, \theta_2)$ | AEP (GWh) | Expected power per turbine (MW) | Exec. time (s) |
|---|---|---|---|---|---|---|
| 81 | 0.5 | 1 | $(5.5\,\text{turb}_{\text{diam}}, 2\,\text{turb}_{\text{diam}}, 73°, -87°)$ | 2860.00 | 4.031 | 4083 |
| 81 | 0.5 | 5 | $(2\,\text{turb}_{\text{diam}}, 3\,\text{turb}_{\text{diam}}, 18°, -80°)$ | 2862.37 | 4.034 | 9611 |
| 81 | 0.5 | 10 | $(2\,\text{turb}_{\text{diam}}, 3\,\text{turb}_{\text{diam}}, 18°, -80°)$ | 2862.76 | 4.035 | 15 981 |
| 81 | 1 | 1 | $(2\,\text{turb}_{\text{diam}}, 6\,\text{turb}_{\text{diam}}, 90°, 70°)$ | 2855.62 | 4.024 | 690 |
| 81 | 1 | 5 | $(2\,\text{turb}_{\text{diam}}, 3\,\text{turb}_{\text{diam}}, 18°, -80°)$ | 2862.29 | 4.034 | 2343 |
| 81 | 1 | 10 | $(2\,\text{turb}_{\text{diam}}, 3\,\text{turb}_{\text{diam}}, 18°, -80°)$ | 2862.29 | 4.034 | 3944 |
| 81 | 2 | 1 | $(2\,\text{turb}_{\text{diam}}, 6\,\text{turb}_{\text{diam}}, 90°, 70°)$ | 2856.23 | 4.025 | 161 |
| 81 | 2 | 5 | $(2\,\text{turb}_{\text{diam}}, 2\,\text{turb}_{\text{diam}}, 18°, -78°)$ | 2857.65 | 4.027 | 399 |
| 81 | 2 | 10 | $(2\,\text{turb}_{\text{diam}}, 2\,\text{turb}_{\text{diam}}, 18°, -76°)$ | 2860.44 | 4.031 | 1211 |
| 100 | 0.5 | 1 | $(2\,\text{turb}_{\text{diam}}, 2\,\text{turb}_{\text{diam}}, 18°, -78°)$ | 3354.24 | 3.829 | 3486 |
| 100 | 0.5 | 5 | $(2\,\text{turb}_{\text{diam}}, 2\,\text{turb}_{\text{diam}}, 18°, -76°)$ | 3356.53 | 3.832 | 9693 |
| 100 | 0.5 | 10 | $(2\,\text{turb}_{\text{diam}}, 2\,\text{turb}_{\text{diam}}, 18°, -76°)$ | 3356.55 | 3.832 | 14105 |
| 100 | 1 | 1 | $(2\,\text{turb}_{\text{diam}}, 2\,\text{turb}_{\text{diam}}, 19°, -78°)$ | 3348.50 | 3.822 | 884 |
| 100 | 1 | 5 | $(2\,\text{turb}_{\text{diam}}, 2\,\text{turb}_{\text{diam}}, 18°, -82°)$ | 3356.21 | 3.831 | 2027 |
| 100 | 1 | 10 | $(2\,\text{turb}_{\text{diam}}, 2\,\text{turb}_{\text{diam}}, 18°, -76°)$ | 3356.55 | 3.832 | 5324 |
| 100 | 2 | 1 | $(2\,\text{turb}_{\text{diam}}, 2\,\text{turb}_{\text{diam}}, 19°, -78°)$ | 3348.57 | 3.823 | 158 |
| 100 | 2 | 5 | $(2\,\text{turb}_{\text{diam}}, 2\,\text{turb}_{\text{diam}}, 18°, -78°)$ | 3354.45 | 3.829 | 746 |
| 100 | 2 | 10 | $(2\,\text{turb}_{\text{diam}}, 2\,\text{turb}_{\text{diam}}, 18°, -76°)$ | 3356.54 | 3.832 | 920 |
| 150 | 0.5 | 1 | $(2\,\text{turb}_{\text{diam}}, 2\,\text{turb}_{\text{diam}}, 18°, -78°)$ | 4372.17 | 3.327 | 3729 |
| 150 | 0.5 | 5 | $(2\,\text{turb}_{\text{diam}}, 2\,\text{turb}_{\text{diam}}, 18°, -78°)$ | 4372.71 | 3.328 | 12013 |
| 150 | 0.5 | 10 | $(2\,\text{turb}_{\text{diam}}, 2\,\text{turb}_{\text{diam}}, 18°, -78°)$ | 4372.71 | 3.328 | 19188 |
| 150 | 1 | 1 | $(2\,\text{turb}_{\text{diam}}, 2\,\text{turb}_{\text{diam}}, 19°, -78°)$ | 4357.54 | 3.316 | 966 |
| 150 | 1 | 5 | $(2\,\text{turb}_{\text{diam}}, 2\,\text{turb}_{\text{diam}}, 18°, -78°)$ | 4372.71 | 3.328 | 4151 |
| 150 | 1 | 10 | $(2\,\text{turb}_{\text{diam}}, 2\,\text{turb}_{\text{diam}}, 18°, -78°)$ | 4372.71 | 3.328 | 4902 |
| 150 | 2 | 1 | $(2\,\text{turb}_{\text{diam}}, 2\,\text{turb}_{\text{diam}}, 19°, -78°)$ | 4357.48 | 3.316 | 339 |
| 150 | 2 | 5 | $(2\,\text{turb}_{\text{diam}}, 2\,\text{turb}_{\text{diam}}, 18°, -78°)$ | 4372.71 | 3.328 | 858 |
| 150 | 2 | 10 | $(2\,\text{turb}_{\text{diam}}, 2\,\text{turb}_{\text{diam}}, 18°, -78°)$ | 4372.71 | 3.328 | 2084 |
| 250 | 0.5 | 1 | $(2\,\text{turb}_{\text{diam}}, 2\,\text{turb}_{\text{diam}}, 90°, 14°)$ | 5241.50 | 2.393 | 462 |
| 250 | 0.5 | 5 | $(2\,\text{turb}_{\text{diam}}, 2\,\text{turb}_{\text{diam}}, 90°, 14°)$ | 5241.50 | 2.393 | 1203 |
| 250 | 0.5 | 10 | $(2\,\text{turb}_{\text{diam}}, 2\,\text{turb}_{\text{diam}}, 90°, 14°)$ | 5241.50 | 2.393 | 1789 |
| 250 | 1 | 1 | $(2\,\text{turb}_{\text{diam}}, 2\,\text{turb}_{\text{diam}}, 90°, 27°)$ | 5202.29 | 2.375 | 262 |
| 250 | 1 | 5 | $(2\,\text{turb}_{\text{diam}}, 2\,\text{turb}_{\text{diam}}, 90°, 14°)$ | 5241.50 | 2.393 | 555 |
| 250 | 1 | 10 | $(2\,\text{turb}_{\text{diam}}, 2\,\text{turb}_{\text{diam}}, 90°, 14°)$ | 5241.50 | 2.393 | 925 |
| 250 | 2 | 1 | $(2\,\text{turb}_{\text{diam}}, 2\,\text{turb}_{\text{diam}}, 90°, 19°)$ | 5150.62 | 2.352 | 100 |
| 250 | 2 | 5 | $(2\,\text{turb}_{\text{diam}}, 2\,\text{turb}_{\text{diam}}, 90°, 18°)$ | 5223.08 | 2.385 | 205 |
| 250 | 2 | 10 | $(2\,\text{turb}_{\text{diam}}, 2\,\text{turb}_{\text{diam}}, 90°, 18°)$ | 5223.08 | 2.385 | 293 |

**Table 2.** Influence of the hyper-parameters $N_\theta$ and $\Delta r$ on the AEP and algorithm A10 execution time.




| Number of turbines | AEP (GW h) | Shape parameters $(r_1, r_2, \theta_1, \theta_2)$ | Expected power / turbine | % wake losses |
|---|---|---|---|---|
| 1 | 42.55 | – | 4.86 MW | 0 |
| 81 | 2862.76 | $(2\,\text{turb}_{\text{diam}}, 3\,\text{turb}_{\text{diam}}, 18°, -80°)$ | 4.03 MW | 17.1% |
| 100 | 3356.55 | $(2\,\text{turb}_{\text{diam}}, 2\,\text{turb}_{\text{diam}}, 18°, -76°)$ | 3.83 MW | 21.2% |
| 150 | 4372.71 | $(2\,\text{turb}_{\text{diam}}, 2\,\text{turb}_{\text{diam}}, 18°, -78°)$ | 3.33 MW | 31.5% |
| 250 | 5241.50 | $(2\,\text{turb}_{\text{diam}}, 2\,\text{turb}_{\text{diam}}, 90°, 14°)$ | 2.39 MW | 50.8% |

**Table 3.** Optimal AEP, shape configuration and mean power per turbine for an increasing number of turbines.

| Number of turbines | AEP (GW h) | Shape parameters $(r_1, r_2, \theta_1, \theta_2)$ | Expected power per turbine | % wake losses | % AEP increase w.r.t. table 2 |
|---|---|---|---|---|---|
| 1 | 42.550 | – | 4.86 MW | 0 | 0 |
| 81 | 2866.697 | $(2\,\text{turb}_{\text{diam}}, 2\,\text{turb}_{\text{diam}}, 88.6°, 17.9°)$ | 4.04 MW | 16.8% | 0.14 % |
| 100 | 3369.709 | $(2\,\text{turb}_{\text{diam}}, 2\,\text{turb}_{\text{diam}}, 88.5°, 17.9°)$ | 3.85 MW | 20.8% | 0.39 % |
| 150 | 4399.174 | $(2\,\text{turb}_{\text{diam}}, 2\,\text{turb}_{\text{diam}}, 88.8°, 17.78°)$ | 3.35 MW | 31.1% | 0.61 % |
| 250 | 5308.184 | $(2\,\text{turb}_{\text{diam}}, 2\,\text{turb}_{\text{diam}}, 87.0°, 17.4°)$ | 2.42 MW | 50.1% | 1.27 % |

**Table 4.** Optimal AEP, shape configuration and mean power per turbine for an increasing number of turbines using a fast heuristic for the optimization of the shape's parameters





## Appendix A: Algorithms in pseudo-code

---

**Algorithm A1** GenerateR1R2$(\Delta r, D_{\min}, D_{\max})$

---

1: $R_{1,2} \leftarrow \emptyset$

2: $r_1 \leftarrow D_{\min}.\text{turb}_{\text{diam}}$

3: angles $\leftarrow \emptyset$

4: **while** $r_1 \leq D_{\max}.\text{turb}_{\text{diam}}$ **do**

5: $\quad r_2 \leftarrow D_{\min}.\text{turb}_{\text{diam}}$

6: $\quad$ **while** $r_2 \leq D_{\max}.\text{turb}_{\text{diam}}$ **do**

7: $\quad\quad R_{1,2} \leftarrow R_{1,2} \cup \{r_1, r_2)\}$

8: $\quad\quad r_2 \leftarrow r_2 + \Delta r$

9: $\quad$ **end while**

10: $\quad r_1 \leftarrow r_1 + \Delta r$

11: **end while**

12: **return** $R_{1,2}$

---

**Algorithm A2** ReduceSearchSpace$(R_{1,2}, \Delta\theta, D_{\min}, D_{\max}, \text{turb}_{\text{diam}}, N_\theta)$

---

1: $\Theta \leftarrow \emptyset$

2: **for** $(r_1, r_2) \in R_{1,2}$ **do**

3: $\quad \Theta_{r_1,r_2} \leftarrow \text{GetBestAngles}(r_1, r_2, \Delta\theta, D_{\min}, D_{\max}, \text{turb}_{\text{diam}}, N_\theta)$

4: $\quad \Theta \leftarrow \Theta \cup \Theta_{r_1,r_2}$

5: **end for**

6: **return** $\Theta$

---





---

**Algorithm A3** GetBestAngles($r_1, r_2, \Delta\theta, D_{\min}, D_{\max}, \mathrm{turb}_{\mathrm{diam}}, N_\theta$)

---

1: farms $\leftarrow \emptyset$

2: $\Theta_{r_1,r_2} \leftarrow \emptyset$

3: $\theta_1 \leftarrow -\frac{\pi}{2} + \Delta\theta$

4: **while** $\theta_1 \leq \frac{\pi}{2}$ **do**

5:     $\theta_2 \leftarrow -\frac{\pi}{2}$

6:     **while** $\theta_2 \leq \theta_1 - \Delta\theta$ **do**

7:         $d_{\mathrm{grid}} \leftarrow \min_{z \in \mathbb{Z}_*^2} \left\| P_{\mathcal{B}_0}^{\mathcal{B}(\theta_1,\theta_2,r_1,r_2)} z \right\|$

8:         **if** $(d_{\mathrm{grid}} \geq D_{\min}.\mathrm{turb}_{\mathrm{diam}}) \wedge (\theta_1 - \theta_2 \leq \pi - \Delta\theta)$ **then**

9:             $n \leftarrow 1$

10:             **for** $i = 0$ **to** $i = 1$ **do**

11:                 **for** $j = 0$ **to** $j = 1$ **do**

12:                     $\mathbf{F}_4(n) \leftarrow P_{\mathcal{B}_0}^{\mathcal{B}(\theta_1,\theta_2,r_1,r_2)} \begin{pmatrix} i \\ j \end{pmatrix}$

13:                     $n \leftarrow n + 1$

14:                 **end for**

15:             **end for**

16:             $\mathrm{aep} \leftarrow \mathbb{E}_W \left[ \mathcal{P}(\mathbf{F}_4, w_s, w_d) \right]$

17:             farms $\leftarrow$ farms $\cup \{(\theta_1, \theta_2, \mathrm{aep})\}$

18:         **end if**

19:         $\theta_2 \leftarrow \theta_2 + \Delta\theta$

20:     **end while**

21:     $\theta_1 \leftarrow \theta_1 + \Delta\theta$

22: **end while**

23: sort(farms) {by decreasing order of aep}

24: **for** $i = 1$ **to** $i = N_\theta$ **do**

25:     $\Theta_{r_1,r_2} \leftarrow \Theta_{r_1,r_2} \cup \{(\mathrm{farms}(i).\theta_1, \mathrm{farms}(i).\theta_2)\}$

26: **end for**

27: **return** $\Theta_{r_1,r_2}$

---



---

**Algorithm A4** GenerateConfigs($R_{1,2}, \Theta, D_{\min}, D_{\max}, \text{turb}_{\text{diam}}$)

---

1: grids $\leftarrow \emptyset$

2: **for** $(r_1, r_2) \in R_{1,2}$ **do**

3:     **for** $(\theta_1, \theta_2) \in \Theta$ **do**

4:        $d_{\text{grid}} \leftarrow \min_{z \in \mathbb{Z}_*^2} \left\| P_{\mathcal{B}_0}^{\mathcal{B}(\theta_1, \theta_2, r_1, r_2)} z \right\|$

5:        **if** $\min_{z \in \mathbb{Z}_*^2} \left\| P_{\mathcal{B}_0}^{\mathcal{B}(\theta_1, \theta_2, r_1, r_2)} z \right\| \geq D_{\min}.\text{turb}_{\text{diam}}$ **then**

6:           grids $\leftarrow$ grids $\cup \{(r_1, r_2, \theta_1, \theta_2)\}$

7:        **end if**

8:     **end for**

9: **end for**

10: **return** grids

---

---

**Algorithm A5** ComputeAllIntersections(grids, $D_{\min}$, $\Omega$)

---

1: intersections_sets $\leftarrow \{\emptyset\}$

2: $n \leftarrow 1$

3: **while** $n \leq \text{card}(\text{grids})$ **do**

4:     {Compute basic shape vector}

5:     $r_1, r_2, \theta_1, \theta_2 \leftarrow \text{grids}(n)$

6:     $v_1 \leftarrow (r_1 \cos(\theta_1), r_1 \sin(\theta_1))$

7:     $v_2 \leftarrow (r_2 \cos(\theta_2), r_2 \sin(\theta_2))$

8:     {Function of the intersections}

9:     $I_{v_1, v_2}(\Delta_1, \Delta_2) := \{k_1 v_1 + k_2 v_2 \in \mathbb{R}^2 \text{ s.t. } [k_1 v_1 + k_2 v_2 + (\Delta_1, \Delta_2)] \in \Omega \, ; \, (k_1, k_2) \in \mathbb{Z}^2\}$

10:     {Compute offset maximising the number of admissible intersections}

11:     $(\Delta_1^*, \Delta_2^*) \leftarrow \arg\max_{\Delta_1, \Delta_2 \in [0,1)^2} \text{card}(I_{v_1, v_2}(\Delta_1, \Delta_2))$

12:     {Add intersections to the set of intersections}

13:     **if** $\text{card}(I_{v_1, v_2}(\Delta_1^*, \Delta_2^*)) \geq N_{\max}$ **then**

14:        intersections_sets $\leftarrow \{(I_{v_1, v_2}(\Delta_1^*, \Delta_2^*), (r_1, r_2, \theta_1, \theta_2))\} \bigcup$ intersections_sets

15:     **end if**

16:     $n \leftarrow n + 1$

17: **end while**

18: **return** intersections_sets

---



---

**Algorithm A6** GreedyInitialization(intersections)

---

1: $(x_0, y_0) \leftarrow \arg\max_{x,y}\{x - y \text{ s.t. } (x, y) \in \text{intersections}\}$

2: $\mathbf{F}_1(1) \leftarrow (x_0 \ \ y_0)$

3: $n_t \leftarrow 1$

4: $\text{intersections} \leftarrow \text{intersections}\setminus\{(x_0 \ \ y_0)\}$

5: **while** $n_t < N_{\max}$ **do**

6: $\quad (x^* \ \ y^*)^{\top} \leftarrow \arg\max_{(x,y)\in\text{intersections}} \mathbb{E}_{\mathrm{W}}\{\mathcal{P}(\mathbf{F}_{n_t} \oplus (x \ \ y), w_s, w_d)\}$

7: $\quad \mathbf{F}_{n_t+1} \leftarrow \mathbf{F}_{n_t} \oplus (x^* \ \ y^*)$

8: $\quad n_t \leftarrow n_t + 1$

9: $\quad \text{intersections} \leftarrow \text{intersections}\setminus\{(x^* \ \ y^*)\}$

10: **end while**

11: **return** $\mathbf{F}_{N_{\max}}$

---

---

**Algorithm A7** LocalSearch($\mathbf{F}_{N_{\max}}$, intersections)

---

1: convergence $\leftarrow \perp$

2: $\text{aep} \leftarrow \mathbb{E}_{\mathrm{W}}\{\mathcal{P}(\mathbf{F}_{N_{\max}}, w_s, w_d)\}$

3: **while** $\neg$ convergence **do**

4: $\quad \mathbf{H}_{N_{\max}} \leftarrow \mathbf{F}_{N_{\max}}$

5: $\quad \text{random\_indices} \leftarrow \text{shuffle}(\llbracket 0, \dots, N_{\max}\llbracket)$

6: $\quad$ **for** indice $\in$ random_indices **do**

7: $\quad\quad$ {compute all possible layouts by moving i$^{\text{th}}$ turbine}

8: $\quad\quad \text{children\_layout} \leftarrow \text{generate\_children}(\mathbf{F}_{N_{\max}}, \text{indice}, \text{intersections})$

9: $\quad\quad$ **for** $(\mathbf{G}_{N_{\max}}, \text{intersections}_G) \in \text{children\_layout}$ **do**

10: $\quad\quad\quad$ **if** $\mathbb{E}_{\mathrm{W}}\{\mathcal{P}(\mathbf{G}_{N_{\max}}, w_s, w_d)\} > \text{aep}$ **then**

11: $\quad\quad\quad\quad \mathbf{F}_{N_{\max}} \leftarrow \mathbf{G}_{N_{\max}}$

12: $\quad\quad\quad\quad \text{aep} \leftarrow \mathbb{E}_{\mathrm{W}}\{\mathcal{P}(\mathbf{F}_{N_{\max}}, w_s, w_d)\}$

13: $\quad\quad\quad\quad \text{intersections} \leftarrow \text{intersections}_G$

14: $\quad\quad\quad$ **end if**

15: $\quad\quad$ **end for**

16: $\quad$ **end for**

17: $\quad$ convergence $\leftarrow \mathbf{F}_{N_{\max}} = \mathbf{H}_{N_{\max}}$

18: **end while**

19: **return** $(\mathbf{F}_{N_{\max}}, \text{aep})$

---





---

**Algorithm A8** generate_children($\mathbf{F}_{N_{\max}}$, indice, intersections)

---

1: children_layout $\leftarrow \emptyset$

2: **for** $(x\ y) \in$ intersections **do**

3: $\quad \mathbf{G}_{N_{\max}} \leftarrow \mathbf{F}_{N_{\max}}$

4: $\quad$ intersection$_G \leftarrow$ intersections

5: $\quad (x_{\text{new}}\ y_{\text{new}}) \leftarrow \mathbf{G}_{N_{\max}}(\text{indice})$

6: $\quad \mathbf{G}_{N_{\max}}(\text{indice}) \leftarrow (x\ y)$

7: $\quad$ intersections$_G \leftarrow$ intersections$_G \setminus \{(x\ y)\} \cup \{(x_{\text{new}}, y_{\text{new}})\}$

8: $\quad$ children_layout $\leftarrow$ children_layout $\cup \{(\mathbf{G}_{N_{\max}}, \text{intersections}_G)\}$

9: **end for**

10: **return** children_layout

---

---

**Algorithm A9** PlaceTurbines(intersections)

---

1: $\mathbf{F}_{N_{\max}} \leftarrow$ GreedyInitialization(intersections)

2: $(\mathbf{F}_{N_{\max}}, \text{aep}) \leftarrow$ LocalSearch($\mathbf{F}_{N_{\max}}$, intersections)

3: **return** $(\mathbf{F}_{N_{\max}}, \text{aep})$

---

---

**Algorithm A10** Aligned_Optimization($\Omega, N_{\max}, N_\theta, D_{\min}, D_{\max}, \Delta r, \Delta \theta$)

---

1: $R_{1,2} \leftarrow$ GenerateR1R2($\Delta r, D_{\min}, D_{\max}$)

2: angles $\leftarrow$ ReduceSearchSpace($R_{1,2}, \Delta \theta, D_{\min}, D_{\max}, \text{turb}_{\text{diam}}, N_\theta$)

3: grids $\leftarrow$ GenerateConfigs($R_{1,2}, \text{angles}, D_{\min}, D_{\max}, \text{turb}_{\text{diam}}$)

4: intersections_sets $\leftarrow$ ComputeAllIntersections(grids, $D_{\min}, \Omega$)

5: layouts $\leftarrow \emptyset$

6: **for** (intersections, $(r_1, r_2, \theta_1, \theta_2)) \in$ intersections_sets {Run in parallel} **do**

7: $\quad (\mathbf{F}_{N_{\max}}, \text{aep}) \leftarrow$ PlaceTurbines(intersections)

8: $\quad$ layouts $\leftarrow$ layouts $\bigcup \{(\mathbf{F}_{N_{\max}}, \text{aep}, \text{intersections}, (r_1, r_2, \theta_1, \theta_2))\}$

9: **end for**

10: layouts $\leftarrow$ sort(layouts) by aep in descending order

11: **return** layouts(0)

---



*Competing interests.* The contact author has declared that none of the authors has any competing interests.

*Copyright statement.* The method described in this publication is the subject of the international patent application number WO 2024/061627.



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
