# Peer review of "Wind farm layout optimization with alignment constraints"

_Wind Energy Science, 2024_

## Referee Comment (RC2)

The reviewer strongly believes that the paper presents critical wind farm layout optimization aspects. The results appear original and very well written.

Page1: Title. While it is generally understood why the author is discussing wind farm layout, it might be clear to revise the title to explicitly mention "offshore" wind farm layout to clarify the study's focus, as offshore constraints differ widely from onshore. This would also improve clarity for future citations.

Page1: Line4. "to the authors' best knowledge" is unnecessary. It is expected that the author performs extensive research and could state that they performed an extensive literature review and did not find any prior studies on the topic.

Page1: Abstract. Generally, improve clarity in the abstract. For instance, when stating, "the contributions of this paper are twofold," explicitly mention the method – heuristics - rather than vaguely referring to it as a method to handle the optimization problem.

Page1: Line13. Provide reference to the claim in Line 13/14

Page3: Line60. $Z^*$ is redundant. Not utilized in the rest of the paper

Page3: Line81. Clarify how the random wind variable was determined. Because the probability distribution of wind speed and direction were introduced on Page 4.

Page4-8: Study domain. Was a regularization term introduced to prevent overfitting specific wind conditions for wake? How will this change the result?

Page5: Line116. $\theta_1$ is a constraint as provided in Fig1, which means the wind farm is constrained to $\theta_1$ between $-90$ and $90$. Are wind farms never aligned outside these angles? See comments on Fig7.

Page5: Line18. $D_{min}$ and $D_{max}$ are not specified as a multiplication of rotor diameter. Some wake models utilize 5D, 6D, 7D etc of the rotor diameter (D)

Page8: Line184. The authors compare their case study to Thomas et al. (2023) DEBO algorithm, but they do not provide a direct numerical comparison of the results

Page14: Line246. The conclusion claims that their proposed method outperforms a baseline but does not specify by what percentage.

General comments:

The heuristic model lacks sufficient detail in the explanation. Because it did not compare with the baseline models to see how they outperformed the others.

Results lack error analysis, confidence intervals, or sensitivity to other wake model assumptions.

Basic explanations for the choice of constraints made are generally not provided. For instance, how does reducing Δr (grid spacing) affect AEP and computational cost?

[Figure]

offshore wind

[revised manuscript text omitted]
}_{\mathrm{diam}}\}$ | $\{6\,\mathrm{turb}_{\mathrm{diam}}\}$ | $\{1°\}$ | $\{1,5,10\}$ | $\left\{\frac{\mathrm{turb}_{\mathrm{diam}}}{2}, \mathrm{turb}_{\mathrm{diam}}, 2\,\mathrm{turb}_{\mathrm{diam}}\right\}$ |

**Table 1.** Value set of each hyper-parameter

210    one can see that the expected power per turbine[1] is growing with respect to $N_\theta$ and decreasing with respect to $\Delta r$. Also, when $N_\theta \geq 5$ and $\Delta r \leq 1\mathrm{turb}_{\mathrm{diam}}$, the expected powers per turbine are similar whatever the value of these hyper parameters. However, as illustrated on fig. 5, the execution time is strongly increasing with respect to $N_\theta$ and strongly decreasing with respect to $\Delta r$. Therefore, if algorithm A10 is run using an AEP computation method more precise and computationally more expensive than the one we used, (see Baker et al. (2021); Thomas et al. (2023)), keeping $N_\theta$ reasonably small ($\approx 5$) and $\Delta r$
215    reasonably large ($\approx 1$) should enable the solving algorithm to find a well-performing layout in a reasonable execution time. In addition, as illustrated in table 1, the solving algorithm often sets the optimal shape parameters $(r_1, r_2)$ to their minimal authorized values. This behavior indicates that the method generates grids with many intersections and, thus, a large degree of freedom for the local search part of the algorithm. The larger the degree of freedom for the local-search algorithm, the better the solution. The best layout for each wind farm size is displayed on fig. 6 and the optimal parameters $(r_1, r_2, \theta_1, \theta_2)$, the optimal
220    AEP, the optimal expected power per turbine, and the execution time are displayed on table 3.

**6    Exploration method's impact on the AEP**

The optimization layout algorithm presented in this paper relies on the discrete exploration of the space of shape parameters $(r_1, r_2, \theta_1, \theta_2) \in [D_{\min}, D_{\max}]^2 \times [-\pi/2, \pi/2]^2$. Despite the angles' search-space reduction technique presented in section 4.2 and algorithm A2, exploring this space using fine discretization is numerically too demanding. Unfortunately, the performance
225    of the optimization depends on the size of the discretization step; the smaller this size, the higher the AEP. Therefore, there is a strong incentive to develop heuristic methods to explore the shape parameters space other than by using the angle search space reduction associated with a brute force-like exploration method. In this section, we present optimization results using such a heuristic to provide a benchmark for an aligned optimization algorithm. Unfortunately, for industrial confidentiality reasons, we do not describe its principle and only focus on the improvement in terms of AEP. Again, we have run the optimization
230    procedure for wind farms of 81, 100, 150, and 250 turbines. The results in terms of optimal shape parameters, AEP, expected power per turbine, and wake losses are summarized in table 4, and the optimal layouts are displayed in fig. 7. The optimal layout obtained using this heuristic exhibits the same behavior as those found in the previous section in terms of $r_1$ and $r_2$. Indeed, these parameters are systemically found to be equal to the lowest possible value. On the contrary, the optimal angles are not the same. One of the alignment directions is conserved ($\approx 18°$), but the other one is quite different even when taking
* * *
[1]For a $N_t$-turbines wind farm, the expected power per turbine is given by the formula $\mathrm{AEP(MWh)}/(8760 \times N_t)$ and allows for comparing wind farms of different sizes.

[Figure]

(a) Expected power per turbines for a 81-turbines wind farm as a function of $N_\theta$ and $\Delta r$

(b) Expected power per turbines for a 100-turbines wind farm as a function of $N_\theta$ and $\Delta r$

(c) Expected power per turbines for a 150-turbines wind farm as a function of $N_\theta$ and $\Delta r$

[Figure]

(d) Expected power per turbines for a 250-turbines wind farm as a function of $N_\theta$ and $\Delta r$

**Figure 4.** Expected power per turbine for 81, 100, 150, and 250 turbines wind farms as a function of the hyper-parameters $N_\theta$, and $\Delta r$.

[Figure]

(a) Execution time of algorithm A10 as a function of the hyper-parameters $N_\theta$, $\Delta r$ for a 81-turbines wind farm

(b) Execution time of algorithm A10 as a function of the hyper-parameters $N_\theta$, $\Delta r$ for a 100-turbines wind farm

(c) Execution time of algorithm A10 as a function of the hyper-parameters $N_\theta$, $\Delta r$ for a 150-turbines wind farm

[Figure]

(d) Execution time of algorithm A10 as a function of the hyper-parameters $N_\theta$, $\Delta r$ for a 250-turbines wind farm

**Figure 5.** Execution time of algorithm A10 for 81, 100, 150, and 250 turbines wind farms as a function of the hyper-parameters $N_\theta$, and $\Delta r$. All optimizations were run on a 12$^{\text{th}}$ Gen Intel(R) i7-12700H 2.30 GHz core.

[Figure]

(a) Best 81-turbines wind farm layout

(b) Best 100-turbines wind farm layout

(c) Best 150-turbines wind farm layout

(d) Best 250-turbines wind farm layout

**Figure 6.** Best layouts for wind farm sizes of 81, 100, 150, 250 turbines using algorithm A10.

[Figure]

[Figure]

235 into account the 180° periodicity of the angles. Concerning the AEP, using a heuristic to explore the space of shape parameters more efficiently allows for improvement. Interestingly, the percentage of AEP increase grows almost linearly concerning the wind farm size and reaches 1% for the larger one. This behavior stems from the decreasing degrees of freedom in the turbine's optimal placing problem as the wind farm size grows. Therefore, for larger farms, the efficiency of the shape parameters optimization algorithm is of greater importance than for smaller farms; thus, there is a more substantial improvement of AEP

240 for large farms when using a better exploration algorithm for the space of shape parameters. These results prove a strong interest in developing efficient heuristics to explore the space of shape parameters.

**7    Conclusions**

This work tackles the wind farm layout optimization problem with alignment constraints. We introduced a model of the corresponding optimization problem and adapted the DEBO algorithm from Thomas et al. (2023) to this new problem. The proposed

245 method is based on an exploration heuristic for computing the grid parameters and a local-search method to place the turbines

Can this statement be quantified. and be explicitly defined in previous section by way of comparison

on the grid's intersections optimally. We have shown that this method performs well on the benchmark of IEA Wind task 37 (Thomas (2022)) by outperforming the initial layout, even though the latter does not satisfy any alignment constraints and is potentially less prone to significant wake losses. Using this numerical example, we have also demonstrated the benefits of developing efficient heuristics for exploring the grid parameters. Indeed, using efficient heuristics allows for a better trade-

[revised manuscript text omitted]

---

## Author Response (AR1)

**Answers to Reviewer #1**

The paper proposes a method to solve the wind farm layout optimization problem while taking into account alignment constraints. These can be relevant when the navigability of vessels within the wind farm is considered. This method is based on an algorithm that parametrizes the possible turbines' positions within the domain through the intersections of a grid based on parallelograms. This enables to reduce the size of the problem and to obtain an effective convergence. The problem is rigorously formulated and the algorithm is widely described within the document. Finally, the selection of the hyperparameters are discussed and the algorithm is used to solve a widely known example to prove its effectiveness. Overall, this work introduces an interesting method to tackle the challenging layout optimization problem, but it could be further improved by making some modifications. Here I have included my suggestions.

GENERAL COMMENTS

The contributions of this paper should be highlighted more within the methodology. This comment mainly refers to the identification of the innovative aspect introduced in this research with respect to the several works available in literature where the turbine layouts are parameterized through the angles and distances of a regular grid. For instance, many studies do not consider the possibility of "not occupying" a grid intersection (which is considered here). Another innovative aspect is the application of a domain composed by multiple regions. Such aspects (along with the other differences) should be highlighted.

We have added the following paragraph on lines 35-38:

*By alignment constraint, we place the turbines at the intersections of a regular grid composed of parallelograms, whose shape and orientation are to be determined, while considering the possibility of not occupying all the grid's intersections. This possibility is a key feature of the proposed algorithm, and its interest is illustrated in the numerical examples presented in the paper*

Overall, the paper defines the optimization problem using clear and rigorous mathematical expressions and definitions. However, a more detailed description of such expressions within the text could facilitate the readability of the work.

The aspect of introducing the alignment constraint to focus for instance on the navigability of vessels within the farm is innovative and interesting. To give additional value to this aspect, I would suggest to add some references where this requirement is mentioned.

The authors of this paper contributed their expertise to industrial partners responding to tenders for offshore wind farm projects. All these projects required alignment constraints, as formulated in this paper. Unfortunately, the technical specifications are not publicly available. However, regular layout constraints are mentioned in two technical reports from the French Ministry of Transport and the Maritime and Coastguard Agency from the UK. These reports are now cited in the introduction on lines 39-40

The fact that the parameters Delta_1 and Delta_2 do not depend on each turbine ensure the alignment constraint. However, it could be interesting to mention (as future work) that allowing small deviations of these parameters for every turbine could be the starting point for a sensitivity analysis based on the relaxation of such constraint.

We have added the following sentence in the conclusion of the paper.

*Finally, to quantify the effect of alignment constraints on the wake losses, one could perform a sensitivity analysis by allowing small displacements of each turbine, resulting in an almost aligned layout.*

Several times the authors refer to the description of the algorithms included in the Appendix. To facilitate the reading, I would suggest to include a description of the algorithm (e.g. using some block visualizations) within the main text.

Done

SPECIFIC COMMENTS

- In the lines 33-34, it is mentioned that there is no method in the literature that takes into account the alignment constraints. However, these are implicitly taken into account when the turbines are placed at the intersection points of a regular grid. Despite the number of optimization variables that define such a grid are limited in most of the studies, this should be mentioned in the introduction.

We have added the following paragraph on lines 35-38:

*By alignment constraint, we place the turbines on the intersections of a regular grid made of parallelograms whose shape and orientation are to be determined while considering the possibility of not occupying all of the grid's intersections. This possibility is a key feature of the proposed algorithm, and its interest is illustrated in the numerical examples of the paper*

- In section 3.1, the parameters used for the parameterization of the grid shape are described only by referring to the Figure 1. However, a brief description within the text could facilitate the reading

We have added the following paragraph on lines 101-103:

*The grid is a parallelogram-based tiling of the plane, the parameters $r_1, r_2$ are the two sides' length of the parallelogram, the parameter $\theta_1$ (resp. $\theta_2$) is the angle formed between the side of the parallelogram of length $r_1$ (resp. $r_2$) and the x-axis. Finally the parameters $v_x, v_y$ is the offset between the origin of the Cartesian and the parallelogram-based grids*

- The mathematical formulation of the objective function (Equation 8) is clear but "over-complicated" with respect to the ones usually present in the literature (even though they are equivalent). I would suggest to provide further description within the text to facilitate the reading.

We have rewritten the definition in a less abstract fashion and including more details to facilitate the reading.

- I would suggest to modify the notation used to indicate the turbine diameter to make the expressions more clear.

Done, the notation is now $D_{\text{turb}}$

- Figures 4 and 5 are quite difficult to understand and interpret. I would suggest to use a 2d visualization including various lines/points for the different parameters.

We have modified these Figures according to the reviewer suggestions.

- In section 5.1 the method used to compute the AEP is described. I would include also the discretization adopted to for the wind speed and the wind direction values, which are relevant for the computational time that is further described.

We included the discretization bins to compute the AEP (line 206)

- In table 2, it is not clear if the computational time column refers to the step 4 of the algorithm described in section 4.1. If this is the case, it would be helpful to mention also the computational time required for the step 2 (section 4.1) as a function of the hyperparameters. Moreover, please highlight why only some combinations of angles are present in the table.

In these tables, the shape parameters displayed are the optimal ones corresponding to the best layout. To avoid any ambiguity, we now name the column "optimal shape parameters."

- In every table that the computational time is mentioned, the processor that has been used should be mentioned to facilitate the reading.

Done

- The title of section 6 does not match exactly its content. I would recommend to modify it in order to enhance that a new method is introduced to increase the performance. Moreover, in this section it is not clear why the focus of the modified algorithm is on the AEP increase instead of the computational time reduction, please provide some arguments. My concern arises since at the beginning you highlight the need of increase the speed of convergence instead of the need of converging to a better result. Finally, I would consider using a visualization to show the increased performance of this method (more effective than a table), e.g. increase of AEP as a function of the number of turbines, also to enhance the linear behavior mentioned in the text.

We have explained in more detail that fine-tuning the shape parameters using the proposed method is too computationally expensive. However, it is necessary to get a good layout AEP-wise. Therefore, an efficient shape-parameter-space exploration method allows for fine-tuning these parameters while keeping computation time reasonable. This explanation ranges from line 243 to 249

**Answers to reviewer #2**

The reviewer strongly believes that the paper presents critical wind farm layout optimization aspects. The results appear original and very well written.

Page1: Title. While it is generally understood why the author is discussing wind farm layout, it might be clear to revise the title to explicitly mention "offshore" wind farm layout to clarify the

study's focus, as offshore constraints differ widely from onshore. This would also improve clarity for future citations.

We have changed the title to "Offshore Wind Farm Layout Optimization with Alignment Constraints".

Page1: Line4. "to the authors' best knowledge" is unnecessary. It is expected that the author performs extensive research and could state that they performed an extensive literature review and did not find any prior studies on the topic.

This useless sentence has been removed.

Page1: Abstract. Generally, improve clarity in the abstract. For instance, when stating, "the contributions of this paper are twofold," explicitly mention the method – heuristics - rather than vaguely referring to it as a method to handle the optimization problem.

The abstract has been modified as follows

This paper makes two contributions. First, we propose a model of AEP maximization with alignment constraints as a mixed-integer nonlinear problem, where the continuous parameters are the parallelogram-based tiling parameters and the discrete variables are the turbines' positions at the tiling's intersections. Second, we provide a heuristic derived from the DEBO algorithm from Thomas et al. (2023) developed by the same team.

Page1: Line13. Provide reference to the claim in Line 13/14

We have added two references, the recent paper on Flowers and the review by Porté-Agel and al. from 2020

- LoCascio, M. J., Bay, C. J., Martinez-Tossas, L. A., Bastankhah, M., and Gorlé, C.: FLOWERS AEP: An Analytical Model for Wind Farm Layout Optimization, Wind Energy, 27, 1563–1580, 2024
- Porté-Agel, F., Bastankhah, M., and Shamsoddin, S.: Wind-Turbine and Wind-Farm Flows: A Review, Boundary-Layer Meteorolgy, 174,1–59, 2020

Page3: Line60. Z* is redundant. Not utilized in the rest of the paper

In fact, $\mathbb{Z}_*$ is used once in the paper in Eq (17) to define the minimal distance constraint. Therefore, we propose to keep this definition in the notations.

Page3: Line81. Clarify how the random wind variable was determined. Because the probability distribution of wind speed and direction were introduced on Page 4.

The whole definition 3 has been rewritten according to reviewer #1 and #2 comments.

Page4-8: Study domain. Was a regularization term introduced to prevent overfitting specific wind conditions for wake? How will this change the result?

This is an excellent question which raises the question of the robustness of the solution to the wind probability distribution. The sensitivity or the objective function with respect to the distribution is easy to compute when the support the probability measure is unchanged, that is to say, when a modified probability writes $\hat{P} := \sum_{n=1}^{N} \big( p(w_s^n, w_d^n) + \Delta p(w_s^n, w_d^n) \big) \delta_{(w_s^n, w_d^n)}$ , where $\sum_n \Delta p(w_s^n, w_d^n) = 0$. In this case the cost variation writes

$$\Delta AEP(F) = \sum_{n=1}^{N} \mathcal{P}(F_n, w_s^n, w_d^n) \Delta p(w_s^n, w_d^n)$$

However, if the support is changed, things are more involved and one could consider using Distributionnally Robust Optimization methods which are out of the scope of the present paper. Nevertheless we have added this problematic as a potential future work in the Conclusion.

Page5: Line116. θ1 is a constraint as provided in Fig1, which means the wind farm is constrained to θ1 between −90 and 90. Are wind farms never aligned outside these angles? See comments on Fig7.

We constraint $\theta_1 \in [-90, 90]$ because of the periodical behavior of the tiling. Indeed, the grid parameterization with $\theta_1 = 110°$ is the same than $\theta_1 = 110 - 180 = -70$. We have added the following sentence in the paper:

*We only consider θ1 ∈ (−π/2, π/2] since θ1 > π/2 (resp. θ1 < −π/2) yields the same tiling as θ1 − π/2 (resp. θ1 + π/2).*

Page5: Line18. Dmin and Dmax are not specified as a multiplication of rotor diameter. Some wake models utilize 5D, 6D, 7D etc of the rotor diameter (D)

We use this parameterization in the numerical examples, but for the sake of readability we chose not to use this parameterization in the definition of the minimal distance.

Page8: Line184. The authors compare their case study to Thomas et al. (2023) DEBO algorithm, but they do not provide a direct numerical comparison of the results

The problem we are facing here is that no other algorithms are dealing with alignment constraints. Therefore, comparing with a layout not satisfying an alignment constraint is unfair since relaxing these constraints allow for a substantial increase in AEP.

Page14: Line246. The conclusion claims that their proposed method outperforms a baseline but does not specify by what percentage.

The increase in AEP with respect to the baseline layout is 0.41% , we have added this figure in the paragraph

General comments:

The heuristic model lacks sufficient detail in the explanation. Because it did not compare with the baseline models to see how they outperformed the others.

We decided to show the increase of performance when using the exploration heuristic without giving details on the exploration method just to illustrate that future works could focus on this part of the overall algorithm since it is possible to both accelerate the algorithm and find better layouts with a good exploration heuristics.

Results lack error analysis, confidence intervals, or sensitivity to other wake model assumptions.

Reviewers #2 and #3 have opposite thoughts on the matter. In the authors' opinion, a sensitivity analysis of the wake models would indeed be interesting, but it should be addressed in a separate

paper that focuses on the robustness of optimization models with respect to uncertainties, including the wake model.

Basic explanations for the choice of constraints made are generally not provided. For instance, how does reducing Δr (grid spacing) affect AEP and computational cost?

We have made numerous changes regarding the impact of these parameters on the AEP and computation time. The 3D plots are now 2D and are much easier to read, better illustrating the effects of these parameters on the algorithm's performance.

**Answers to reviewer #3**

The article tackles a relevant and interesting topic in wind energy: alignment constraints imposed on wind farm developers by maritime authorities to secure the navigation of boats near wind farms. Despite being well-written from the mathematical point of view, in my view, the article needs improvements in all the bullet points described below.

Line 18: Many other reviews have come up since Herbert-Acero.

We have added more recent reviews.

Line 25: "12" should be written as "twelve".

Done

Introduction: I didn't understand the criteria for this literature review. For instance, why are genetic algorithms and particle swarms even mentioned? Did you do anything related? The idea of randomly mentioning a few papers does not seem scientifically sound. I would stick to papers that are at least more or less related to the general topic. What are the papers that were the closest to analyzing alignment constraints? For instance, line 39 would align well with Fischetti's work cited in line 24.

It is true that our optimization method does not belong to the same category as genetic algorithms. However, all of these papers produce aligned layouts. This is no prerequisite, but a consequence of their modelization using a coarse orthonormal discretization of the admissible domain. Therefore, we decided to include these references in the bibliography. The paper by Fischetti, describes the problem at hand as a MINLP problem. This model allows for a finer discretization of the admissible domain which can output wind farms which do not satisfy any alignment constraint.

In the Introduction, it is not clear (or not easy to identify) what benchmark the authors refer to. Is this the benchmark by Thomas et al. (2023)? What are you benchmarking? AEP?

The benchmark is indeed the numerical example treated in this paper.

Line 71 and 72: I tried the search for "conjonction" in some dictionaries, but I am suspicious there is a typo mistake in there.

We have corrected this mistake

Figure 1 legend: there should be an endpoint on that sentence (and all the other subfigures).

We have corrected this mistake

Section 5.2: why do you need to use Dmax? Not clear.

Dmax can be chosen as large as one needs. In practice it is convenient to limit this value to avoid running Algorithm 5 for shape parameters such that the number of admissible intersections is inferior to Nmax. We have added the following explanation in the algorithm's description

*There is no upper limit to the Dmax value. However, it is useless to set this parameter at a large value. Indeed, beyond a certain value d > Dmin, any shape parameters configuration such that r1, r2 > d will produce coarse grids with less than Nmax admissible intersections.*

Figure 2. This figure seems a bit raw and could be further improved. Probably, at the very least, with the names of each zone. The size can be reduced. It is right now occupying unproportional space in the article.

We have deleted the figure since the zones are well visible in the numerical results illustrations.

The whole section 5.2 talks about the influence of hyper-parameters on the AEP. How does that relate to your storylines? Aren't you showcasing your methodology for alignment constraints? Shouldn't you only have sections that support that? I didn't understand the point of this section and how that supports the storyline.

This reviewer has an opinion opposite to reviewer #2. The authors of this paper agree with the reviewer's point of view and focus on the optimization algorithm rather than the hyperparameters of wake models. However, the hyperparameters we refer to are the optimization algorithm's hyperparameters. Therefore, we analyze the effect of these tuning parameters on the performance of the optimization algorithm in terms of the objective function (AEP) and computation time.

Line 227: what kind of heuristics? I get that it is confidential, I just don't get why are you showing it in this open-access scientific publication. No one can replicate/confirm/compare the results.

We chose to present these results to illustrate the strong incentive for improving the exploration method to enhance the algorithm's performance.

Line 249: not demonstrated (the benefits of heuristics).

The benefits are showcased on Table 4 and mainly consist in finding a layout with a better AEP.

Line 250: The wind community cannot use the heuristics; it is proprietary.

The purpose is to illustrate that future works should focus on shape parameter exploration techniques.

In my opinion, the paper should focus on showcasing the methodology for the alignment constraints. I think it is well contextualized in that sense, as lines 35-38 describe. Other than that, I couldn't understand how all the hyperparameter analysis contributed to advancing the state of the art in the topic.